# Canine respiratory coronavirus in Thailand undergoes mutation and evidences a potential putative parent for genetic recombination

Panida Poonsin,[1,2] Vorapun Wiwatvisawakorn,[3] Jira Chansaenroj,[4] Yong Poovorawan,[4] Chutchai Piewbang,[1,2] Somporn Techangamsuwan[1,2]

**ABSTRACT**   Canine respiratory coronavirus (CRCoV) is associated with canine infectious respiratory disease complex. Although its detection has been reported worldwide, the genomic characteristics and evolutionary patterns of this virus remain poorly defined. In this study, 21 CRCoV sequences obtained from dogs in Thailand during two episodes (2013–2015, group A; 2021–2022, group B) were characterized and analyzed. The genomic characteristics of Thai CRCoVs changed from 2013 to 2022 and showed a distinct phylogenetic cluster. Phylogenetic analysis of the spike (S) genes divided the analyzed CRCoV strains into five clades. The full-length genome characterization revealed that all Thai CRCoVs possessed a nonsense mutation within the nonstructural gene located between the S and envelope genes, leading to a truncated putative nonstructural protein. Group B Thai CRCoV strains represented the signature nonsynonymous mutations in the S gene that was not identified in group A Thai CRCoVs, suggesting the ongoing evolutionary process of Thai CRCoVs. Although no evidence of recombination of Thai CRCoV strains was found, our analysis identified one Thai CRCoV strain as a potential parent virus for a CRCoV strain found in the United States. Selective pressure analysis of the hypervariable S region indicated that the CRCoV had undergone purifying selection during evolution. Evolutionary analysis suggested that the CRCoV was emerged in 1992 and was first introduced in Thailand in 2004, sharing a common ancestor with Korean CRCoV strains. These findings regarding the genetic characterization and evolutionary analysis of CRCoVs add to the understanding of CRCoVs.

**IMPORTANCE**   Knowledge of genomic characterization of the CRCoV is still limited and its evolution remains poorly investigated. We, therefore, investigated the full-length genome of CRCoV in Thailand for the first time and analyzed the evolutionary dynamic of CRCoV. Genomic characterization of Thai CRCoV strains revealed that they possess unique genome structures and have undergone nonsynonymous mutations, which have not been reported in previously described CRCoV strains. Our work suggests that the Thai CRCoVs were not undergone mutation through genetic recombination for their evolution. However, one Thai CRCoV strain PP158_THA_2015 was found to be a potential parent virus for the CRCoV strains found in the United States. This study provides an understanding of the genomic characterization and highlights the signature mutations and ongoing evolutionary process of CRCoV that could be crucial for monitoring in the future.

**KEYWORDS**   canine respiratory coronavirus, evolutionary dynamic, genome characterization, phylogenetic analysis, Thailand

Address correspondence to Somporn Techangamsuwan, somporn.t@chula.ac.th.

The authors declare no conflict of interest.

See the funding table on p. 13.

Coronaviruses (CoVs) have been known to infect humans and a wide range of animals, including domestic, exotic, and wild species (1–5). Regarding the diversity of CoVs and their potential infection in various hosts, as they use similar receptors for viral entry (6–8), interspecies transmission of CoVs has been a concern reported on several occasions (9–12). According to the International Committee on Taxonomy of Viruses classification (ICTV) in 2020, the CoVs that can infect humans and animals have been divided into four genera: *Alphacoronavirus*, *Betacoronavirus*, *Gammacoronavirus*, and *Deltacoronavirus* (13, 14). Animal CoVs have been identified in various domestic hosts including cattle, swine, horses, poultry, and companions. Among them, there are several CoVs that are commonly found in cats and dogs, including feline enteric coronavirus, feline infectious peritonitis virus, canine enteric coronavirus (CCoV), canine respiratory coronavirus (CRCoV), and pantropic CCoV (15–17). Concerning interspecies transmission, as recent evidence indicates infection in humans of the CoV that arose from dogs (18–20), and dogs live close to their owners, the genetic relationship and evolutionary pattern of CoVs in dogs must be determined.

CRCoV is an enveloped, single-stranded, positive-sense RNA virus that belongs to the family *Coronaviridae*, genus *Betacoronavirus*, and subgenus *Embecovirus*. CRCoV is genetically close to bovine coronavirus (BCoV) and human coronavirus OC43 (HCoV-OC43) (21). The genome of CRCoV is approximately 30 kb in length (22, 23). The genomic organization of CRCoV comprises two main open reading frames (ORFs), ORF1ab and 1a, which encompass approximately two-thirds of the whole genome; structural genes that include hemagglutinin esterase (HE), spike (S), envelope (E), membrane (M), and nucleocapsid (N) genes; and an accessory gene that encodes nonstructural (NS) proteins located between the S and E genes (24). Similar to other coronaviruses, the S gene preserves the hypervariable region and is used to differentiate either coronaviral species or variants (25–27). Although CRCoV is a CoV that can cause disease in dogs, it is different from CCoV, which is associated with enteric disease in infected dogs because of the dissimilarity in nucleotide and amino acid levels (28).

Betacoronaviruses share a close genetic relationship with each other (29–31); however, the relationship between CRCoV and other betacoronaviruses is still in question, regardless of the experimental infection of BCoV in puppies (32). There is evidence that HCoV-OC43 shares a common ancestor with BCoV (33). A CRCoV strain K37 from Korea has been found to be a recombinant strain of the BCoV strain Quebec from Canada and CRCoV strain BJ232 from China (23). Although there has been one report of successful BCoV inoculation in puppies (32), the evolution and origin of CRCoV are still unclear.

CRCoV is one of the etiological agents associated with canine infectious respiratory disease complex (34). CRCoV infection affects the early stage of the canine infectious respiratory disease complex and subsequently causes severe respiratory distress due to secondary infection from other pathogens, either bacteria or viruses (35). CRCoV was first discovered in kennel dogs showing respiratory distress in the United Kingdom in 2003 (28). Since then, CRCoV has been detected in dogs worldwide, and the prevalence of CRCoV has been reported in various countries, including the United Kingdom, Italy, Sweden, Japan, China, South Korea, and New Zealand (22–24, 28, 36–38). On the other hand, CRCoV antibodies have been identified in a large number of dog sera in Europe, South Korea, and New Zealand (34, 39, 40), indicating widespread exposure to CRCoV. Despite the detection of CRCoV being broadly reported, only a few studies have focused on genomic analysis and genetic evolution, resulting in limited information regarding the genetic characterization of CRCoV.

Although CRCoV has been identified in a large number of dog populations worldwide and in Thailand since 2014 (41, 42), the genomic characterization of CRCoV has not been fully investigated, and few CRCoV sequences have been characterized. Together with various reports of CRCoV, scant full-length genomic characterizations of the virus have been available, and the evolutionary dynamics of CRCoVs remain to be elucidated. In the present study, we aimed to characterize the full-length CRCoV genomes of CRCoVs found

in Thailand. To expand our window of investigation, full-length genomes of Thai CRCoVs that were obtained from positive samples collected from two episodes (2013–2015 and 2021–2022) were compared, and the evolution of CRCoV genomes was analyzed.

## RESULTS

### Genomic characterization of CRCoV genome

A total of 21 CRCoV complete coding sequences (group A, $n$ = 8; group B, $n$ = 13) from this study were successfully characterized. Among them, the obtained Thai CRCoV strains had high nucleotide identity (99.6%–100%). Specifically, the Thai CRCoV strains from dogs in group A had the highest identity (99.7%–99.8%) to the CRCoV strain BJ232 (KX432213.1) from China, whereas the Thai CRCoV strains in group B had the highest nucleotide identity (99.6%–99.7%) to the CRCoV isolate USA 4 (ON133848.1) (Table S3).

The CRCoV group A sequences had a 30,579-bp coding sequence length. The first two-thirds of the genome consisted of ORF1ab (nt 65–21,348) and ORF1a (nt 65–13,216), followed by 32 kDa NS (nt 21,358–22,194), HE (nt 22,206–23,480), S (nt 23,495–27,586), E (nt 28,262–28,516), M (nt 28,531–29,223), and N (nt 29,233–30,579). Three NS genes were identified between the S and E genes as 4.9 kDa (nt 27,576–27,673), 2.7 kDa (nt 27,741–27,818), and 12.8 kDa (nt 27,946–28,275), respectively. Within CRCoV group B, sequences had a 30,569-bp nucleotide length that contained genetic segments and nucleotide positions similar to those of CRCoV group A. However, the group B CRCoV sequences had a 10-nt deletion at 61st to 70th positions of the 4.9 kDa NS gene. This deletion was not found in group A CRCoV strains, resulting in the length being 10 nt shorter than that of the group A CRCoV strains. Interestingly, all 21 complete CRCoV coding sequences from Thailand showed a nucleotide deletion at the 11th nt position of the 4.9 kDa NS gene that was not present in other previously described CRCoV sequences (Fig. 1A). Furthermore, a nonsense mutation was identified at the 88th nt position of the 4.9 kDa NS gene, resulting in translation of putative 3.58 and 3.08 kDa NS protein in group A and group B CRCoV, respectively, instead of 4.9 kDa NS protein (Fig. 1B).

Regarding the S gene analysis, the pairwise nucleotide identity among Thai CRCoV sequences showed 99.3%–100% identity. Similar to the analysis results based on a nearly complete genome, the group A Thai CRCoV strains shared the highest nucleotide identity with CRCoV strain BJ232 (KX432213.1) from China identified in 2014, ranging from 99.8% to 99.9%, whereas the group B Thai CRCoV strains had the highest nucleotide identity to CRCoV isolate CRCoV21032451-2 (OQ351919.1) detected in China in 2021, ranging from 99.7% to 99.9%. All Thai CRCoVs shared the lowest nucleotide identity with CRCoV isolate USA 1 (ON133845.1), detected in the United States in 2021, with 98.1%–98.6% nucleotide identity (Table S4). These findings reflect what we found in the analysis based on nearly complete coding sequences.

Several amino acid changes were noticed in the S genes of Thai CRCoV strains. Notably, most of the amino acid changes were located in the S1 subunit. In addition, three amino acid signatures were found in all strains of group B that were located at positions L96R, F281V, and M407I. These amino acid changes were found only in group B Thai CRCoV strains, and one previously published CRCoV sequence identified in 2021 from China (OQ351919.1).

### Phylogenetic analysis

Due to the limited complete genome sequences of CRCoV in the GenBank database, we categorized clades of CRCoV based on the S gene. In total, 37 CRCoV strains were segregated into five clades (Fig. 2). The first clade consisted of CRCoV sequences found in Japanese strain 02/005, AB242262.1; Italian strain 240/05, EU999954.1; Swedish strain CRCoV1_Sweden, LR721664.1; and some CRCoV strains (isolates USA 1, ON133845.1; USA 2, ON133846.1; and USA 3, ON133847.1) identified in the USA. The other CRCoV strain found in the USA (isolate USA 5, ON133844.1) was categorized into the second unique clade. CRCoV strain T101 (AY150272.1) and strain 4182 (DQ682406.1) found in the United

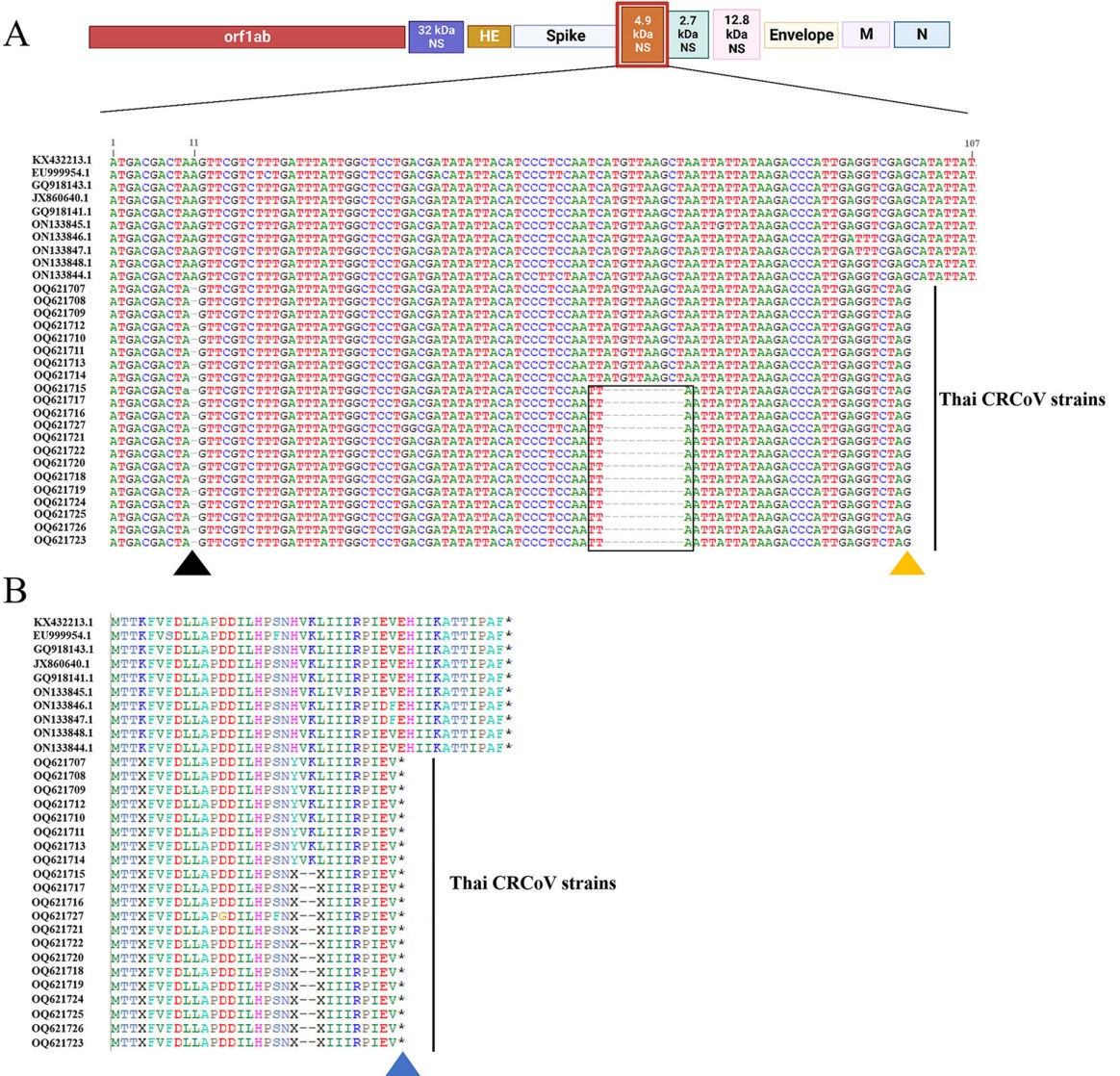

**FIG 1** Molecular analysis of 4.9 kDa NS genes of Thai CRCoVs and reference strains. The nucleotide position of Thai strains represented one nucleotide deletion (black arrowhead) and a nonsense mutation (yellow arrowhead). The 10-nucleotide deletion in Thai strains from 2021 to 2022 is highlighted in the black box (A). The deduced amino acid sequence alignment indicated a truncated amino acid in Thai strains (blue arrowhead) (B).

Kingdom were grouped into the third clade with CRCoV strain 06/075 (AB370269.1) detected in Japan. The fourth clade included all CRCoV strains found in South Korea. The Thai CRCoV strains formed a unique clade (the fifth clade) together with two CRCoV strains from China (strain BJ232, KX432213.1 and CRCOV21032451-2, OQ351919.1) and one strain from the USA (strain USA 4, ON133848.1). This clade was further divided into three subclades, 5a, 5b, and 5c, which were supported by high bootstrap values. The group A Thai CRCoV strains were grouped into subclade 5b, whereas the group B Thai CRCoV strains were clustered into subclade 5c.

The topologies of Thai CRCoVs based on nearly complete coding genomes, HE, M, and N genes were revealed to be in concordance with the phylogenetic tree constructed from the S gene (Fig. 3). Notably, the phylogenetic relationships of CRCoV isolates USA 2 and 3 showed discordant results as these strains were included in the same cluster as Thai CRCoVs in the topologies based on M and N genes (Fig. 3C and D), but they were grouped distantly in the phylogenetic trees of the S gene, nearly complete coding

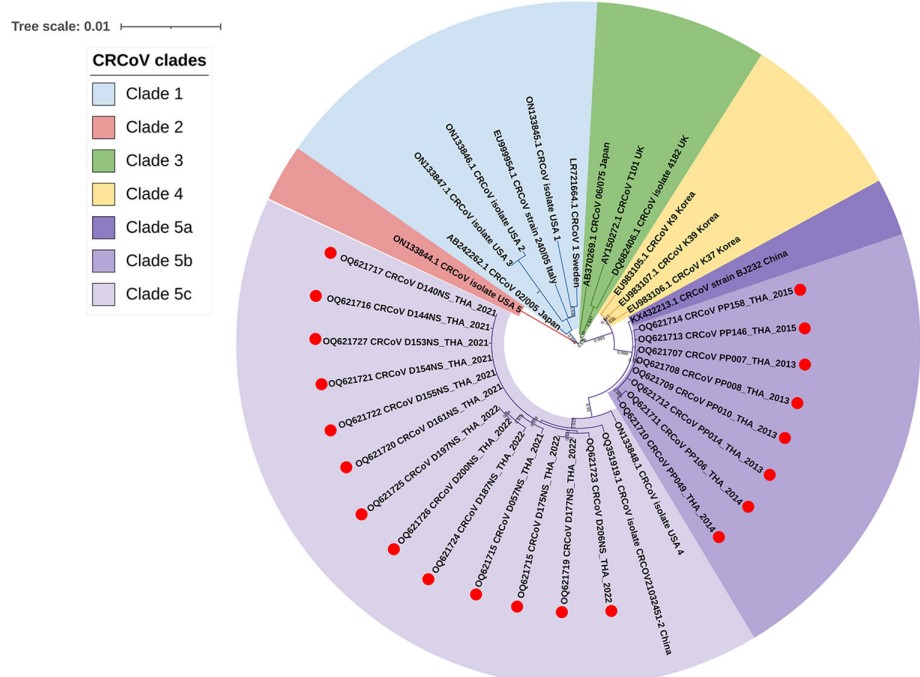

**FIG 2** Phylogenetic tree of CRCoV strains based on S gene sequences obtained from this study using the maximum likelihood method. The five CRCoV clades are marked by different colors. The red dots indicate the CRCoV sequences from this study. Thai CRCoVs were grouped into the fifth clade, which was further divided into three subclades. Subclade 5a contained one sequence from China (CRCoV strain BJ232, KX432213.1). The group A Thai CRCoV strains were grouped into subclade 5b, whereas the group B Thai CRCoV strains were clustered into subclade 5c together with CRCoV isolate USA 4 (ON133848.1) and CRCoV isolate CRCOV21032451-2 (OQ351919.1) from China.

sequences, and HE gene (Fig. 2, 3A and B). These inconsistent results suggested the possibility that CRCoV isolates USA 2 and 3 were recombinant viruses.

## Recombination analysis

Recombination analysis of Thai CRCoV strains with other CRCoVs indicated that none of the Thai CRCoV strains had recombined with other CRCoV strains. However, we found that CRCoV isolate USA 3 (ON133847.1) was a potential recombinant virus that had the CRCoV isolate USA 1 (ON133845.1) and CRCoV isolate PP158_THA_2015 (OQ621714) as putative major and minor parents, respectively. The recombination breakpoint was identified within the intersecting region between the 12.8 kDa NS, E, M, and N genes (Fig. 4A), which was supported by statistical methods including Recombination Detection Program (RDP, GENECONV, BootScan, MaxChi, Chimaera, SiScan, and 3Seq, with *P* values of $8.967 \times 10^{-15}$, $1.495 \times 10^{-14}$, $1.163 \times 10^{-3}$, $7.003 \times 10^{-6}$, $3.673 \times 10^{-7}$, $1.353 \times 10^{-10}$, and $7.731 \times 10^{-11}$, respectively. Similarity plots and bootscan analyses also supported the RDP results (Fig. S1; Fig. 4B). The results showed that the recombinant CRCoV isolate USA 3 (ON133847.1) had a high nucleotide identity to the CRCoV isolate USA 1 (ON1338475.1; blue line). However, in the region of the 12.8 kDa NS, E, M, and N genes, the CRCoV isolate USA 3 had nucleotide identity to the CRCoV isolate PP158_THA_2015 (OQ621714; red line). The putative recombinant breakpoint of the CRCoV isolate USA 3 was located at nt 27,827–30,710 (Fig. 4).

## Evolutionary analysis of CRCoV

A data set of S genes of CRCoVs was used to estimate the divergence timeline and evolutionary history of CRCoV. The maximum clade credibility revealed that Thai CRCoVs formed a distinct clade (Fig. 5). The Tamura-Nei 93 (TN93) model with a log-normal

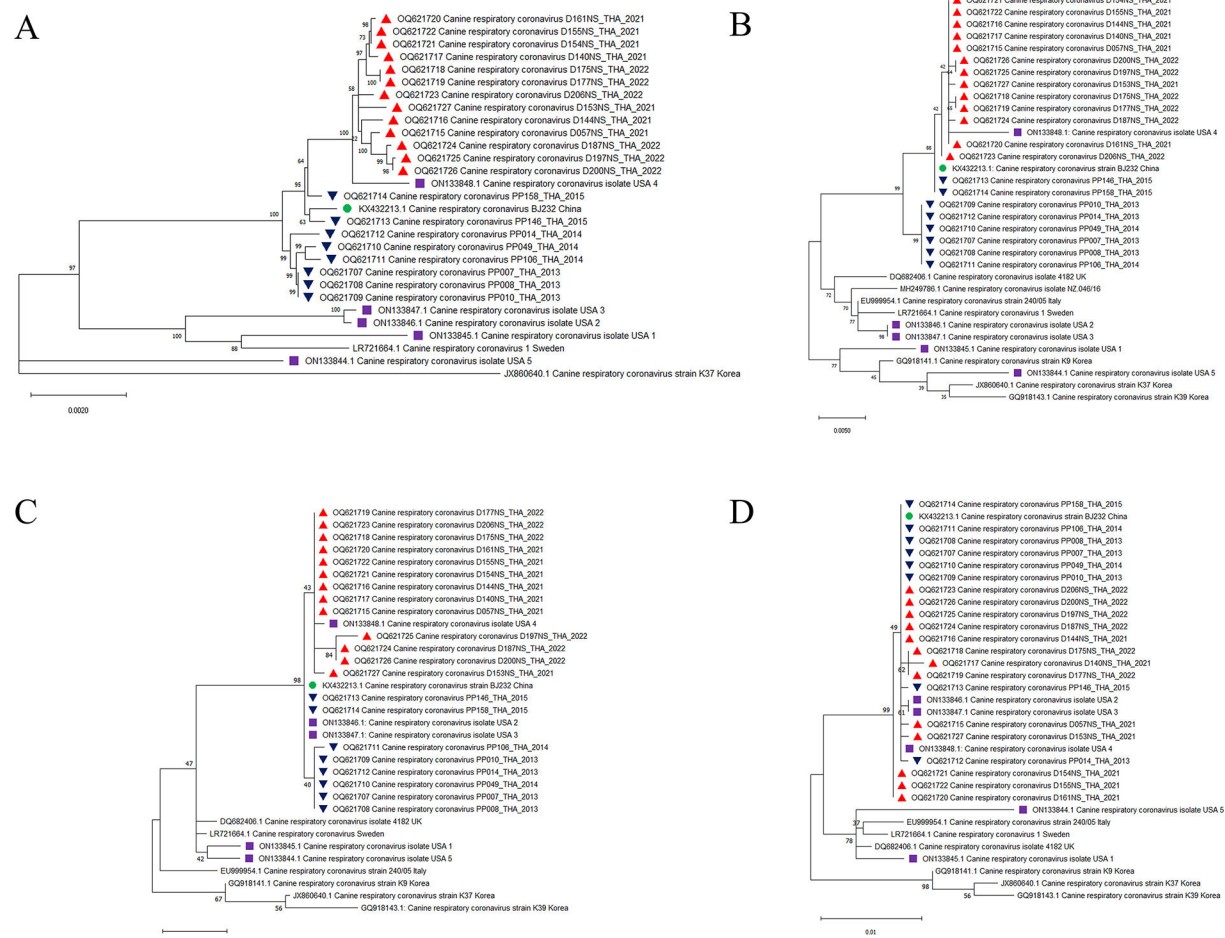

**FIG 3** The phylogenetic trees of CRCoV strains based on the nearly complete coding genomes (A), HE (B), M (C), and N (D) genes. Symbols of different colors indicate Thai CRCoV strains from 2013 to 2015 (▼), Thai CRCoV strains from 2021 to 2022 (▲), CRCoV strain BJ232 from China (●), and strains from the USA (■). The best-fit substitution model for the phylogram constructed based on the nearly complete coding genome is general time reversible with gamma distribution and invariant sites (GTR + G + I), Tamura-Nei 93 with gamma distribution and invariant sites (TN93 +G + I) for HE and S genes, and Tamura 3-parameter with gamma distribution and invariant sites (T92 +G + I) for the M and N genes. All sequences were analyzed using the maximum likelihood method with 1,000 bootstrap replicates.

relaxed clock model was used as a suitable best-fit substitution model for aligned sequences. The overall evolutionary rate was estimated to be $5.246 \times 10^{-4}$ substitutions/site/year [95% highest posterior density (HPD): $3.8265–7.052 \times 10^{-4}$]. CRCoV shared the most common ancestor with BCoV, and the estimated time to the most recent common ancestor of CRCoV was approximately 1992 (95% HPD: 1988–1996). However, we found that CRCoV was first introduced to Thailand in 2004 (95% HPD: 2003–2006) and shared a common ancestor with the Korean CRCoV strains. Meanwhile, group B Thai CRCoV strains shared a common ancestor with CRCoV isolate USA 4 presented in 2017 (95% HPD: 2016–2019).

## Selective pressure analysis

The dN and dS ratios from the S gene alignment of CRCoV were derived from the mixed-effects model of evolution (MEME) and single likelihood ancestor counting (SLAC) methods. The overall dN/dS ratios indicated that the S gene of CRCoV has undergone negative selection pressure (dN/dS <1). However, there were potential positive selection sites of S genes found in the analysis using the MEME, FEL, and Fast Unconstrained Bayesian AppRoximation (FUBAR) methods (Table S2). The MEME test showed the

A

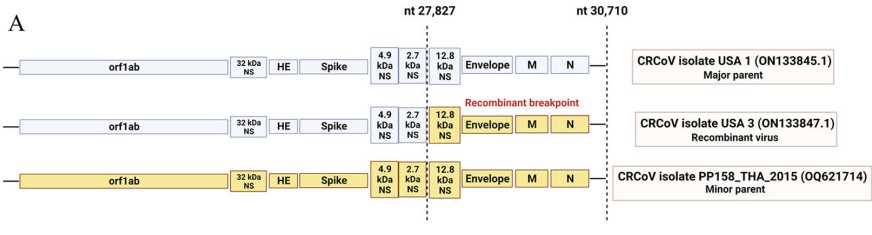

B

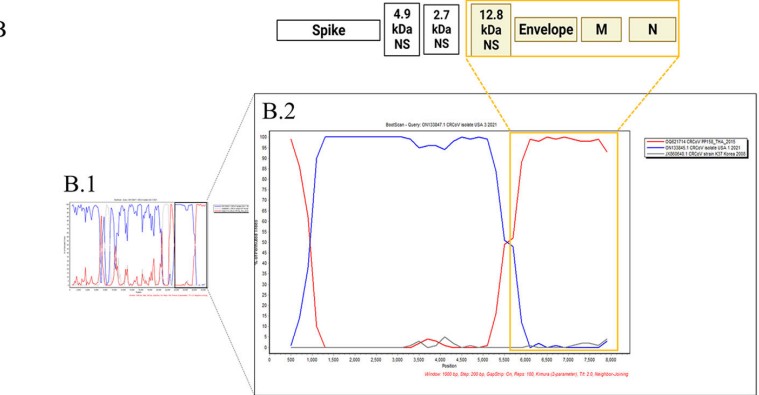

**FIG 4** The schematic of the recombinant CRCoV isolate USA 3 (ON133847.1) genome organization. CRCoV isolate USA 1 (ON133845.1; blue box) and CRCoV isolate PP158_THA_2015 (OQ621714; yellow box) served as putative major and minor parents, respectively. The recombination breakpoint is located at nt 27,827–30,710 (A). Bootscan analysis based on the nearly complete genome (B.1) and bootscan analysis based on the S gene, accessory genes, E, M, and N genes (B.2) of the recombinant CRCoV isolate USA 3 (ON133847.1) as a query and compared with CRCoV isolate PP158_THA_2015 (OQ621714; red line), CRCoV isolate USA 1 (ON133845.1; blue line), and CRCoV strain K37 Korea (JX860640.1; gray line). The y-axis indicates the permuted trees. A window size of 1,000 bp and a step size of 200 bp were used for bootscan analysis (B).

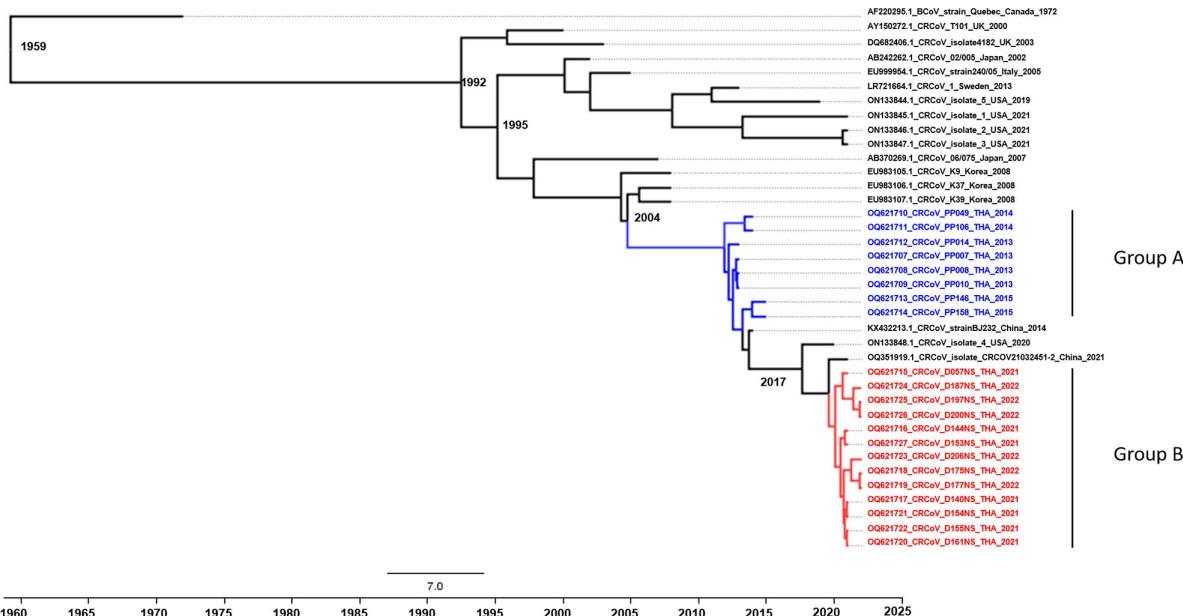

**FIG 5** The phylogenetic tree with timeline estimated divergences based on 37 complete S genes of CRCoV sequences. The phylogeny includes 21 CRCoV sequences from Thailand. The blue line represents group A Thai CRCoV sequences, and the red line represents group B Thai CRCoV sequences. The S gene sequence of BCoV was used as an outgroup sequence. The years of the most recent common ancestor are shown in the nodes.

episodic positive diversifying selection sites of the S gene at codons 74 and 432, whereas the FEL and FUBAR methods showed that codon 1,239 of the S gene was a potentially positive selected site.

## DISCUSSION

Although the occurrence of CRCoV has been reported in many countries (22, 23, 28, 34, 36–38, 42), the molecular characteristics of the virus are not well studied and remain to be elucidated. To date, only three complete coding CRCoV sequences found in three countries have been reported (22, 23, 38). Studies investigating CRCoV through genetic characterization and phylogenetic analysis in other countries are, therefore, needed to indicate the evolutionary origin and roles of genomic mutation of the virus. In this study, we aimed to investigate the genomic characterization of complete coding sequences of the CRCoVs found in Thailand and subsequently compared them with other CRCoV strains. Evolutionary analysis of the S gene of the CRCoV was also studied. To see the evolutionary dynamics more clearly, we selected two CRCoV-positive cohort samples: The first cohort was obtained from 2013 to 2015 (group A), and the second cohort was obtained from 2021 to 2022 (group B). Based on the characterization of the complete coding sequences of Thai CRCoVs, this study demonstrated that the nucleotide identity of group A Thai CRCoV strains is most genetically similar to CRCoV strain BJ232 from China. By contrast, group B Thai CRCoV strains were genetically related to CRCoV isolate USA 4 (ON133848.1). We also found that the Thai CRCoVs contained a unique genome in the region of accessory genes between the S and E genes. We identified a premature stop codon in the 4.9 kDa NS gene that resulted in all CRCoV Thai strains having a truncated 4.9 kDa NS protein. We speculated that this might have resulted from the accessory gene being dispensable for replication of the virus (43) and that they lost these nucleotides during the evolution process (1, 44). Similar to the findings for CCoV, ORF3 was found in CCoV-I, but it appears as a remnant in CCoV-II (45). This observation might be related to changes in viral tropism. Like BCoV, the strains associated with respiratory disease in cattle have nucleotide deletions in this region, but they were not found in the BCoV strains associated with enteric diseases (46). Furthermore, nucleotide deletions causing premature stop codons in ORF3c of the feline enteric coronavirus alter the tropism from enterocytes to monocytes, resulting in the emergence of pathogenic feline infectious peritonitis virus (47, 48). However, the relationship between the effects of nucleotide deletion of the CRCoV found in this study and tropism changes could not be determined based on current information, and this should be further investigated. Instead of representing the 4.9 kDa NS gene as in other CRCoV strains, CRCoV strain 4182 UK presents different nucleotides in this accessory gene due to a deletion of two nucleotides before a stop codon of the 4.9 kDa NS protein that resulted in the absence of the stop codon and the presence of an additional 12 amino acids, leading to the translation of the 8.8 kDa NS protein (21). It is possible that CRCoV strain 4182 UK arose from different ancestors with other CRCoV strains (49). However, other CRCoV reference strains discovered after CRCoV strain 4182 UK presented the 4.9 kDa NS gene with no nucleotide insertions or deletions. Therefore, the presented truncated 4.9 kDa NS protein of Thai CRCoV strains in this study is assumed to be the outcome of the CRCoV evolution in that they might have lost or split accessory genes during the CRCoV evolution (44). However, the function of this accessory gene in CRCoV is still unknown. Therefore, it is recommended that further studies investigate the function of this accessory gene.

Analysis of the S genes showed that the Thai CRCoV strains presented signature nonsynonymous mutations, which consisted of L96R, F281V, and M407I. These mutations were found only in group B, which consisted of Thai CRCoV sequences retrieved from 2021 onward. In group A, this pattern of mutation was not observed. This finding might indicate that group B CRCoVs are the predominant strains in Thailand; however, intensive genomic characterization in larger-scale investigations is needed to verify our speculation. Furthermore, the CRCoV isolate CRCOV21032451-2 identified in 2021 from China also had these mutations, whereas the CRCoVs identified in 2019 to 2021 from the

United States did not have these mutations. It could be speculated that these mutations might be related to the ongoing evolution of CRCoV in Asia. Interestingly, we found that all these amino acid changes were located in the S1 subunit of the S gene, which is a hypervariable region and is also considered an important region associated with CRCoV infection.

The maximum likelihood phylogenetic analysis based on S genes divided CRCoV strains into five clades, and all Thai CRCoV strains were grouped into the fifth clade. Evolutionary analysis indicated that group A Thai CRCoV strains shared a common ancestor with group B Thai CRCoV strains. Furthermore, the phylogram based on the nearly full-length genome indicated that no group A Thai CRCoV strains were included in subclade 5c, which only contained group B Thai CRCoV strains. These findings suggest that Thai CRCoV strains have undergone mutation for 10 years. However, these data must be interpreted with caution because of the limited CRCoV sequences. Further research should be done to investigate the genome characterization of CRCoV in other countries.

CoVs have undergone mutation via genetic recombination that leads to the emergence of new coronavirus variants (50–53). Previously, the CRCoV strain K37 identified in South Korea was considered to be a recombinant strain, and its recombination event occurred in the S gene (23). In this study, no recombination event was found in the Thai CRCoV strains, which might indicate that the Thai CRCoV strains did not undergo recombination processes in their evolution. However, we found that one Thai CRCoV strain (CRCoV isolate PP158_THA_2015) served as the potential parental strain of the recombinant CRCoV isolate USA 3. This finding may indicate the possibility that Thai CRCoVs might have been introduced to the USA recently. However, due to the scarcity of complete genomes of CRCoV available in the GenBank database, recombination analysis could have failed to identify recombination events if all parental strains share the same recombination pattern. Recombination analysis based on a larger scale of CRCoV sequences is, therefore, necessary to confirm our speculation.

Evolutionary analysis of the CRCoV S genes indicated that the mutation rate of this virus is in the range of the mutation rates found in other coronaviruses, including BCoV and HCoV-OC43 (51, 54). Interestingly, CRCoV was first discovered in 2003 in the United Kingdom (28), and one retrospective study reported finding two CRCoV-positive dogs in 1996 (55). However, the molecular clock analysis of CRCoV using S genes in the present study indicated that the time to the most recent common ancestor of CRCoV was estimated in 1992. Nevertheless, the Thai CRCoV samples in this study were selected during only two different timeframes. Further investigations of the CRCoV sequences obtained through a longitudinal study should be included for analysis to increase more precise time to the most recent common ancestor. Due to the fact that the evolutionary analysis of the CRCoV in this study was limited to low numbers of available CRCoV sequences in the database, the evolutionary analysis presented in this study may generate inaccurate results. Further evolutionary analysis based on larger numbers of CRCoV sequences is needed and will provide more accurate results.

The high variability of S genes, including the identified mutation of Thai CRCoV in this study, could be an important turning point in CRCoV evolution. To determine whether environmental forces affect the mutation of CRCoV in S genes, selective pressure analysis was performed using different methods. The results showed that the S codons were under purifying selection, and three signature amino acid mutations in the S genes of Thai CRCoVs appeared as negatively selected sites. This finding implies that CRCoV appears to undergo mutations without the effect of external forces. Thus, these mutations were the outcome of the evolutionary process by the virus more than the effect of the environmental forces that shape the genotype of the viral difference. However, we identified two positive sites (74 and 432) in the S gene using the MEME method and one positive site (1239) using the FEL and FUBAR methods. Positive selection sites in the S region have been previously reported in BCoV (56) and HCoV-OC43, and most of the positively selected sites were mapped to the N-terminal domain, which is important in the transmission process of betacoronavirus (57). The positive

selection sites in the S protein identified in the present study might imply that mutations in the S protein of CRCoV can influence host receptor binding or result in the escape of host immunity, which needs to be further investigated.

In conclusion, this study obtained the genomic characterization and evolutionally analyzed the complete coding genome of CRCoV in Thailand for the first time. We found that Thai CRCoV strains are unique and have undergone mutations, which could indicate the evolution process found for CRCoV. Although we did not find that genetic recombination forced evolution in the Thai CRCoV strains, one CRCoV strain found in this study might become a potential parent virus for the CRCoV found in the USA. Therefore, further observations focusing on CRCoV characterization, genetic recombination, and evolutionary analysis are essential and should be considered in future studies.

## MATERIALS AND METHODS

### RNA extraction and CRCoV detection

The 21 fresh-frozen nasal swab samples were subjected to viral RNA extraction using the QIAamp Viral RNA Mini Kit (Qiagen, Hilden, Germany) according to the manufacturer's instructions. Briefly, 140 µL of swab supernatant was added to 560 µL of Buffer AVL containing 5.6 µL of carrier RNA. The mixtures were subsequently subjected to RNA extraction using the automated extractor QIAcube connect (Qiagen, Hilden, Germany). The obtained viral RNA quantity and purity were evaluated by spectrophotometric analysis using a Nanodrop Lite (Thermo Fisher Scientific Inc., Waltham, MA, USA). The extracted viral RNA was then stored at −80°C until assayed.

To repeatedly confirm the presence of CRCoV RNA in the samples that previously tested positive on RT-PCR (41), the extracted viral RNA samples were subjected to CRCoV detection using one-step RT-PCR with newly designed primers targeting 351 bp of the S gene. The primers used in this assay were designed based on the alignment of previously published CRCoV sequences available in the GenBank database and subsequently validated *in silico* based on the melting temperature and hairpin structure via the OligoAnalyzer Tool (https://sg.idtdna.com/calc/analyzer). The designed primers used for CRCoV detection are listed in Table S1. Briefly, 2 µL of extracted RNA was used as a template and subsequently mixed with OneStep RT-PCR master mix (Qiagen, Hilden, Germany) containing 5 µL of 5XQIAGEN OneStep RT-PCR buffer (containing 12.5 mM MgCl$_2$), 1 µL of dNTP mix (containing 10 µM of each dNTP), 0.6 µM of each primer, 1 µL of QIAGEN OneStep RT-PCR Enzyme Mix, and RNase-free water, with a total volume of 25 µL. The cycling amplification process consisted of incubation at 50°C for 30 min for the RT process, then 95°C for 15 min for the initial PCR activation step, followed by 40 cycles of the initial denaturation step at 95°C for 30 s, 58°C for 30 s for the annealing process, 72°C for 45 s for the extension process, and a final elongation step at 72°C for 10 min. The thermocycling was performed using QIAamplifier 96 (Qiagen, Hilden, Germany).

Amplified PCR products were visualized and analyzed by automated capillary gel electrophoresis via the QIAxcel Advanced System (Qiagen, Hilden, Germany). Briefly, amplified PCR products were run using the QIAxcel DNA Screening Kit (Qiagen, Hilden, Germany). The samples were run simultaneously with a QX DNA size marker of 100 bp– 2.5 kb and a QX alignment marker of 15 bp/3 kb. The assay was run at 5 kV of injection voltage, with 10 s for the injection time, and 6 kV of separation voltage, with 320 s for the separation time (AM320 method) and then analyzed via QIAxcel ScreenGel software. The PCR product samples that presented CRCoV-targeted amplicons were subsequently subjected to a next-generation sequencing-based method (Celemics, South Korea) to confirm the presence of the CRCoV RNA.

### Full-length genome characterization of CRCoV

The CRCoV-positive RNA samples that tested positive on the RT-PCR screening test were subjected to full-length genome characterization. Briefly, the extracted RNA samples were subjected to complementary DNA construction using SuperScript III Reverse

Transcriptase (Invitrogen, Life Technologies, Carlsbad, CA, USA), following the manufacturer's protocol. The synthesized complementary DNA samples were then subjected to full-length genome characterization using PCRs with multiple sets of primers that were designed from the CRCoV alignment as described above. The primers used for full-length genome characterization are described in Table S1. PCRs were performed in a total volume of 25 µL of GoTaq Green Master Mix (Promega, Madison, WI, USA), according to the manufacturer's recommendation. The reactions consisted of a mixture of 12.5 µL of 2× GoTaq Green Master Mix (*Taq* DNA polymerase, dNTPs, MgCl$_2$, and reaction buffers), 0.3 µM of each forward and reverse primer, and 1 µL of complementary DNA template. Thermocycling conditions were 95°C for 5 min of the initial denaturation step followed by 40 cycles of 95°C for 1 min, 54–60°C according to the optimal annealing temperature of each primer pair (Table S1) for 1 min, and 72°C for 1 min and 30 s, with a final extension step of 72°C for 10 min. The PCR products were visualized and analyzed using automated capillary gel electrophoresis as protocols and the instruments described above. The positive amplicons were sent out for genetic sequencing using a next-generation sequencing-based method.

## Genetic and phylogenetic analysis of CRCoV

All Thai CRCoV sequencing fragments obtained from each PCR and primer sets were assembled into contiguous sequences using BioEdit v7.0.5.3. The assembled sequences were then aligned with CRCoV sequences available in the GenBank database using the MAFFT algorithm (58). The nucleotide sequences of all Thai CRCoV strains were compared to the reference sequences of CRCoV using pairwise identity matrix software embedded in BioEdit. The obtained complete coding genome sequences of CRCoV found in this study were submitted to the GenBank database under accession numbers OQ621707-OQ621727. Phylogenetic trees were constructed based on the nearly complete genome sequences of CRCoV and individual gene sequences (HE, S, M, and N genes), as additional trees for further analysis, using MEGA X (59). The substitution model for the constructed phylogram was selected by the software using the Find Best DNA/Protein Models (ML) option. To categorize CRCoV clades, the phylogenetic tree obtained based on the S gene was visualized and labeled using iTOL version 6.0 (https://itol.embl.de/).

## Recombination analysis

To identify any potential recombination events among CRCoV strains, multiple recombination detection methods, including RDP, GENECONV, BootScan, MaxChi, Chimera, SiScan, and 3Seq, provided in the RDP4 v4.101 were used (60). The positive detection of recombination signals with *P* values less than 0.01 in at least four out of the seven methods was considered to indicate potential recombinant strains (61). Potential recombinant strains were further confirmed by similarity plots and bootscan analyses using the SimPlot software package v3.5.1. The putative recombinant was set as a query and run with a window of 1,000 nucleotides and 200 nucleotide steps (23).

## Genetic evolutionary analysis of the CRCoV S gene

The evolutionary analysis of CRCoV was performed to identify the origin of CRCoV. The alignment of various CRCoV S genes was used as a template. Briefly, a data set of 37 CRCoV S sequences and one BCoV S sequence was used to perform an evolutionary analysis using the Bayesian Markov chain Monte Carlo model, implemented in BEAST v2.4.8 (62). A jModelTest (63) was performed to identify the best-fitting nucleotide substitution model for multiple alignment sequences. The best-fit substitution model under a log-normal relaxed and strict clock model with constant population sizes as priors were implemented to account for varied evolutionary rates among lineages. The Coalescent Bayesian Skyline tree prior and empirical base frequencies were implemented under the best-fit clock model and subsequently run for 100 million chains, sampling

every 10,000th generation, with the first 10% discarded as burn-in. The convergence of parameters was confirmed by calculating the effective sample size using the TRACER program v1.7.0 (64). Maximum clade credibility trees were annotated using TreeAnnotator v1.8.3 (62). A phylogenetic tree with estimated divergence, a variable timeline, posterior probability, and the 95% HPD was generated and displayed using FigTree v1.4.3.

## Selective pressure analysis

To determine whether nucleotide substitution mutation resulted from the rapid adaptation of CRCoV, selective pressure analysis was performed on the CRCoV nucleotide coding sequence using the Datamonkey Adaptive Evolution Server (65–67). Due to the fact that the S gene of the CRCoV is considered the most hypervariable region, selective pressure tests were performed on the CRCoV S gene. Nonneutral selection of nucleotide substitutions was calculated based on the ratio of nonsynonymous (dN) to synonymous (dS) substitutions per site using the maximum likelihood phylogenetic reconstruction with the general reversible nucleotide substitution model. The nonneutral selection was executed using different models, including the SLAC, FEL, and mixed-effects model of evolution methods and Fast Unconstrained Bayesian AppRoximation (FUBAR), which were implemented using the HyPhy software package in the Datamonkey server. Statistical significance was set at $P \leq 0.1$ for SLAC, FEL, and MEME methods. The FUBAR method was run with a posterior probability of 0.9. A Bayes factor of 50 was used to estimate the rates of dN and dS within an individual codon. Codons were predicted to be under positive diversifying selection, neutral mutations, and negative selection whether the posterior probabilities were dN/dS >1, dN/dS = 1, and dN/dS <1, respectively.

## ACKNOWLEDGMENTS

P.P. received a grant from The Thailand Research Fund through the Royal Golden Jubilee Ph.D. Program (Grant No. NRCT5-RGJ63001-013) and The Second Century Fund (C2F), Chulalongkorn University. C.P. is supported by the Ratchadapisek Somphot Fund for Postdoctoral Fellowship, Chulalongkorn University. Y.P. is supported by the Center of Excellence in Clinical Virology of Chulalongkorn University/King Chulalongkorn Memorial Hospital (GCE 59009-30-005). J.C. is supported by the The Second Century Fund (C2F), Chulalongkorn University. S.T. is partly supported by the National Research Council of Thailand (NRCT): R. Thanawongnuwech NRCT Senior Scholar 2022 #N42A650553. This research is funded by Thailand Science Research and Innovation Fund Chulalongkorn University (CU_FRB65_hea(89)_194_31_13) (to S.T.) and the 90th Anniversary of Chulalongkorn University Fund (Ratchadaphiseksomphot Endowment Fund) (to P.P.).

The authors declare that there is no conflict of interest regarding the publication of this article.

## AUTHOR AFFILIATIONS

[1]Department of Pathology, Faculty of Veterinary Science, Chulalongkorn University, Bangkok, Thailand
[2]Animal Virome and Diagnostic Development Research Unit, Faculty of Veterinary Science, Chulalongkorn University, Bangkok, Thailand
[3]SLV Pet Hospital, Bangkok, Thailand
[4]Department of Pediatrics, Faculty of Medicine, Center of Excellence in Clinical Virology, Chulalongkorn University, Bangkok, Thailand

## AUTHOR ORCIDs

Somporn Techangamsuwan  http://orcid.org/0000-0002-4888-2677

Microbiology Spectrum

## FUNDING

| Funder | Grant(s) | Author(s) |
|--------|----------|-----------|
| National Research Council of Thailand (NRCT) | N42A650553 | Somporn Techangamsuwan |
| Chulalongkorn University (CU) | GCE 59009-30-005 | Yong Poovorawan |
| Chulalongkorn University (CU) | | Panida Poonsin |
| Ratcahdapisek Somphot Fund for Postdoctoral Fellowship, Chulalongkorn University | | Chutchai Piewbang |

## AUTHOR CONTRIBUTIONS

Panida Poonsin, Data curation, Formal analysis, Investigation, Methodology, Software, Validation, Visualization, Writing – original draft | Vorapun Wiwatvisawakorn, Resources | Jira Chansaenroj, Formal analysis, Software, Validation | Yong Poovorawan, Supervision | Chutchai Piewbang, Conceptualization, Formal analysis, Methodology, Project administration, Supervision, Validation, Visualization, Writing – review and editing | Somporn Techangamsuwan, Conceptualization, Funding acquisition, Project administration, Resources, Supervision, Visualization, Writing – review and editing

## DATA AVAILABILITY

Twenty-one full-length genome sequences of CRCoV were submitted to the NCBI database under GenBank accession numbers OQ621707–OQ621727.

## ETHICS APPROVAL

This study was approved by the Institutional Animal Care and Use Committee (No. 2231001) and the Institutional Biosafety Committee (No. 2131020) of Chulalongkorn University. In all, 21 nasal swab samples that tested positive on routine CRCoV reverse-transcription (RT)-PCR (41) collected from dogs with respiratory diseases from 2013 to 2015 (group A, $n = 8$) and 2021 to 2022 (group B, $n = 13$) were retrieved from the archive at the Department of Pathology, Faculty of Veterinary Science, Chulalongkorn University. All collected samples were subsequently subjected to further investigation.

## ADDITIONAL FILES

The following material is available online.

### Supplemental Material

**Tables S1 to S2, Fig. S1 (Spectrum02268-23-s0001.docx).** Primer list, selective pressure analysis, SimPlot analysis.
**Table S3 (Spectrum02268-23-s0002.xlsx).** Complete coding sequence identity of the Thai CRCoV strains.
**Table S4 (Spectrum02268-23-s0003.xlsx).** Spike sequence identity of the Thai CRCoV strains.

### Open Peer Review

**PEER REVIEW HISTORY (review-history.pdf).** An accounting of the reviewer comments and feedback.

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
