## [Reviewer comments · Microbiology Spectrum]

Microbiology Spectrum

Canine respiratory coronavirus in Thailand undergoes mutation and evidences a potential putative parent for genetic recombination

Panida Poonsin, Vorapun Wiwatvisawakorn, Jira Chansaenroj, Yong Poovorawan, Chutchai Piewbang, and Somporn Techangamsuwan

Corresponding Author(s): Chutchai Piewbang, Chulalongkorn University Faculty of Veterinary Science

Review Timeline:

Submission Date:

June 6, 2023

Accepted:

July 27, 2023

Editor: Frederick S. Kibenge

Reviewer(s): Disclosure of reviewer identity is with reference to reviewer comments included in decision letter(s). The following individuals involved in review of your submission have agreed to reveal their identity: Anastasia N. Vlasova (Reviewer #1)

Transaction Report:

DOI: <https://doi.org/10.1128/spectrum.02268-23>

July 27, 2023

Dr. Chutchai Piewbang
Chulalongkorn University Faculty of Veterinary Science
Pathology
Henri-Dunant
Pathumwan, Bangkok
Thailand

Re: Spectrum02268-23 (Canine respiratory coronavirus in Thailand undergoes mutation and evidences a potential putative parent for genetic recombination)

Dear Dr. Chutchai Piewbang:

Your manuscript has been accepted, and I am forwarding it to the ASM Journals Department for publication. You will be notified when your proofs are ready to be viewed.

Sincerely,

Frederick S. Kibenge
Editor, Microbiology Spectrum
